# How does discharge against medical advice affect risk of mortality and unplanned readmission? A retrospective cohort study set in a large UK medical admissions unit

Anand Alagappan [1], Thomas J G Chambers,[2,3] Erik Brown,[4] Sheila M Grecian,[3] Khalida Ann Lockman[1,5]

AA and TJGC are joint first authors.

For numbered affiliations see end of article.

**Correspondence to**
Dr Thomas J G Chambers;
tom.chambers@ed.ac.uk

## ABSTRACT

**Objectives** To assess the frequency of discharge against medical advice (DAMA) in a large UK teaching hospital, explore factors which increase the risk of DAMA and identify how DAMA impacts patient risk of mortality and readmission.

**Design** Retrospective cohort study.

**Setting** Large acute teaching hospital in the UK.

**Patients** 36 683 patients discharged from the acute medical unit of a large UK teaching hospital between 1 January 2012 and 31 December 2016.

**Measurements** Patients were censored on 1 January 2021. Mortality and 30-day unplanned readmission rates were assessed. Deprivation, age and sex were taken as covariates.

**Results** 3% of patients discharged against medical advice. These patients were younger (median age (years) (IQR)): planned discharge (PD) 59 (40–77); DAMA 39 (28–51), predominantly of male sex (PD 48%; DAMA 66%) and were of greater social deprivation (in three most deprived quintiles PD 69%; DAMA 84%). DAMA was associated with increased risk of death in patients under the age of 33.3 years (adjusted HR 2.6 (1.2–5.8)) and increased incidence of 30-day readmission (standardised incidence ratio 1.9 (1.5–2.2)).

**Limitations** Readmission to acute hospitals outside of the local health board may have been missed. We were unable to include information regarding comorbidity or severity of presentation.

**Conclusions** These data highlight the vulnerability of younger patients who DAMA, even in a free-at-the-point-of-delivery healthcare setting.

## STRENGTHS AND LIMITATIONS OF THIS STUDY

⇒ This is the first study of which we are aware in which mortality and readmission rates are assessed in a 'free-at-the-point of delivery' acute medical unit.

⇒ We adjusted mortality and readmission rates for age, deprivation and sex in a dataset of 26 187 discharges from the acute medical unit.

⇒ We were not able to include information about morbidity or severity of presentation at time of index admission.

⇒ We show that discharge against medical advice is associated with an increase in adjusted risk of mortality and unplanned readmission, independent of the direct financial burden of healthcare.

⇒ Younger patients who discharge against medical advice highlight a vulnerable patient cohort who may benefit from intensified medical, psychological and social intervention to reduce mortality and demand for acute medical services.

## INTRODUCTION

Accounting for an estimated 1%–2% of hospital discharges in the USA,[1–4] discharge against medical advice (DAMA) occurs when a patient chooses to leave hospital prior to recommendation from their medical team. Of concern, the Nationwide Inpatient Sample in the USA highlighted that the annual prevalence of DAMA has increased by 1.9%/annum between 2002 and 2011.[2] These figures are not unique to the USA; studies in other parts of the world show comparable figures although we could not identify examples in the literature from UK-based cohorts.[5–9]

Several factors associate with risk of DAMA; with a difference in perceived recovery (compared with the physician) and financial constraints thought to be most likely.[2] DAMA is prevalent among vulnerable individuals, particularly those with a pre-existing diagnosis of mental health disorders and substance misuse.[10 11]

Patients choosing to leave the hospital against the advice of senior clinicians are at increased risk of significant adverse outcomes. For example, in previous studies in the USA and Canada, patients who DAMA are up to 13.4% more likely to be readmitted within 30 days and up to 0.6% more likely to

do die within 30 days.[10 11] This is thought to be related to a combination of them missing important diagnostic tests and treatment but may also reflect that underlying sociological, economic and psychological vulnerability contribute to decision making around discharge.

## Aims

We aimed to establish the contemporary incidence of DAMA in a large acute UK teaching hospital in a free-at-the-point-of-delivery healthcare system, identify demographic factors associated with DAMA and to explore if DAMA was associated with risk of readmission and mortality. We considered that such data would be essential to frame revision of policies regarding the optimal management of patients choosing to DAMA.

## METHODS

The study was approved by the local Caldicott guardian.

### Study population

We conducted a retrospective analysis of all patients who were discharged from the acute medical unit (AMU) at the Royal Infirmary of Edinburgh (RIE) between 1 January 2012 and 31 December 2016. The RIE serves as the largest local hospital for the region of Lothian, UK which has a population of 900 000. The RIE has a mean annual admission rate of 60 000/year and covers a geographical area of 700 square miles. The AMU is the admissions unit through which all adult medical admissions are handled. Patients are either treated and discharged directly or admitted downstream to their appropriate specialty if longer courses of treatment are required.

Data were compiled from electronic patient records; at each discharge, nursing staff document if the discharge was planned, against medical advice or was a mortality.

### Outcomes and follow-up

The outcome measures were readmission (within 30 days) and mortality up until the point of censor on 1 January 2021.

### Data extraction

Data were collected using TRAK (Oracle) information gathering software. The Information Services Division of NHS Lothian provided the data linkage between the demographics and morbidity and mortality. Demographic variables included were sex, age and Scottish Index of Multiple Deprivation (SIMD). Readmission dates were obtained from the electronic health record and mortality data obtained from linkage with National Records of Scotland.[12] 3.8% of patients did not have an SIMD as no fixed address of residence was associated with the admission. These reflect patients with no fixed abode and for simplicity, we included them among patients in the most deprived quintile. Age strata were defined taking into account that standardised mortality crossed unity at 36 years. We defined three strata giving cutoffs of 33.3 and 66.6 years. Patients admitted from the accident and emergency department following accidental or intentional overdose are cohorted into a toxicology bay in the AMU and are given a specific toxicology code. We thus coded patients as either 'general medicine' or 'toxicology'. The STROBE checklist for cohort studies was followed to ensure necessary items have been reported.

Readmissions have been identified as any patient who has an unplanned inpatient stay anywhere in NHS Lothian, in the 30 days following discharge from AMU. Inpatient stays are only counted against the most recent previous stay so as not to count consecutive stays. Inpatient stays that may have occurred outside of NHS Lothian have not been captured.

Statistical analysis was conducted using *R* V.4.0.0. Raw data are presented with comparisons between DAMA and planned discharge groups assessed using Pearson's $\chi^2$ test for discrete variables and Mann Whitney U-test for continuous variables. Multiple variable logistic regression was used to assess risk factors for DAMA. Cox proportional hazards modelling for mortality was completed using the *survival* package.[13] For each covariate, we tested proportionality of hazards using Pearson product-moment correlations of Schoenfeld residuals with time: we stratified age, SIMD and length of stay to minimise loss of proportionality. We chose to stratify SIMD class as <4 and ≥4 as the Lothian population is skewed to less deprived quintiles. Length of stay was stratified as <2 days or >2 days given the left skew of the data (predominant short admissions). We included stratification of the Cox models by year of discharge given the trend for reducing numbers of discharges over the study period. Multiple variable logistic regression was used to assess odds of DAMA and cause of death. Indirect standardised incidence rates were calculated using the *sir* function of the *popEpi* package.[14] Significance was taken as p<0.05. Data are presented as medians with IQR.

### Patient and public involvement

Patients or the public were not involved in the design, conduct or reporting or dissemination plans of our research

## RESULTS

There were 37 409 discharges from the AMU between 1 January 2012 and 31 December 2016 inclusive. We excluded 13 patients for whom we had no date of birth. A total of 10 918 discharges were excluded as they were not the first discharge for each patient within the study period; 300 were excluded as the patient had died during the admission. A total of 26 187 first patient discharges were assessed. Of these, there were 25 430 planned discharges and 757 DAMA (3%) (figure 1).

Patients who DAMA were more likely to be male (48.1% of PD population vs 65.8% of the DAMA population ($X^2$=92.4, p<0.0001); younger (median age 60 years (PD) vs 39 years (DAMA); p<0.0001(Mann-Whitney U test); of more deprived SIMD ($\chi^2$=167.5, p<0.0001);

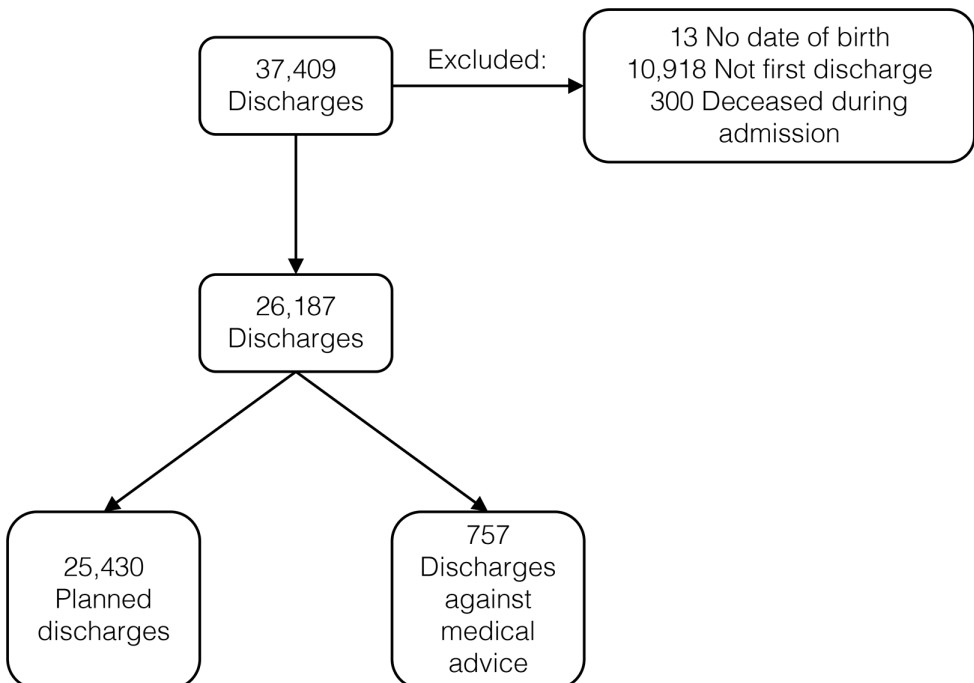

**Figure 1** Flowchart showing the patients excluded from the initial 37 409 discharges included in initial analyses.

were more likely to have an inpatient stay of under 2 days (96.3% (PD) vs 99.3% (DAMA); $X^2$=19.4, p<0.0001) and were more likely to be discharged from the toxicology unit (18.7% (PD) vs 37.5% (DAMA); $\chi^2$=168.1, p<0.0001) (table 1). The number of discharges from AMU reduced for each study year but the proportion of DAMA did not vary significantly ranging between 2.4% and 3.1%.

**Table 1** Summary of differences in patients with PD compared with those who DAMA

|  | PD (N=25 430) | DAMA (N=757) | Total (N=26 187) | P value |
|---|---|---|---|---|
| Sex |  |  |  | <0.001 |
| Male (%) | 12 228 (48.1%) | 498 (65.8%) | 12 726 (48.6%) |  |
| Age at admission (years) |  |  |  | <0.001 |
| Median (IQR) | 59 (40–77) | 39 (28–51) | 59 (40–76) |  |
| SIMD |  |  |  | <0.001 |
| 1 most deprived | 6817 (27.2%) | 310 (43.1%) | 7127 (27.7%) |  |
| 2 | 5746 (23.0%) | 183 (25.4%) | 5929 (23.0%) |  |
| 3 | 4765 (19.0%) | 112 (15.6%) | 4877 (18.9%) |  |
| 4 | 4001 (16.0%) | 70 (9.7%) | 4071 (15.8%) |  |
| 5 least deprived | 3697 (14.8%) | 45 (6.2%) | 3742 (14.5%) |  |
| Length of stay<2 days |  |  |  | <0.001 |
| <2 days | 22 058 (86.7%) | 721 (95.2%) | 22 779 (87.0%) |  |
| >2 days | 3372 (13.3%) | 36 (4.8%) | 3408 (13.0%) |  |
| Specialty |  |  |  | <0.001 |
| General medicine | 20 626 (81.1%) | 473 (62.5%) | 21 099 (80.6%) |  |
| Toxicology | 4804 (18.9%) | 284 (37.5%) | 5088 (19.4%) |  |
| Age of death, years |  |  |  | <0.001 |
| n | 5734 | 68 | 5802 |  |
| Median (IQR) | 84 (75–90) | 59 (49–70) | 84 (75–90) |  |

Difference assessed by $\chi^2$ for categorical values, or Mann Whitney U test for continuous variables.
DAMA, discharge against medical advice; PD, planned discharge; SIMD, Scottish Index of Multiple Deprivation.

**Table 2** Multiple variable logistic regression for characteristics and their association with odds of discharge against medical advice

| Characteristic | OR | 95% CI | P value |
|---|---|---|---|
| **Sex** | | | |
| Female | — | — | |
| Male | 1.90 | 1.63, 2.23 | <0.001 |
| **SIMD 2 level** | | | |
| <4 most deprived | — | — | |
| ≥4 least deprived | 0.56 | 0.45, 0.68 | <0.001 |
| **Stratified age (years)** | | | |
| 13–33.3 | — | — | |
| 33.34–66.6 | 0.67 | 0.56, 0.79 | <0.001 |
| >66.6 | 0.11 | 0.08, 0.15 | <0.001 |
| **Specialty** | | | |
| General medicine | — | — | |
| Toxicology | 1.39 | 1.17, 1.65 | <0.001 |
| **LOS** | | | |
| ≤2 days | — | — | |
| >2 days | 0.37 | 0.25, 0.51 | <0.001 |

SIMD, Scottish Index of Multiple Deprivation.

Age under 33.3 years, male sex, a more deprived SIMD, discharge from a toxicology admission and, length of stay <2 days were all significantly associated with increased odds of DAMA as assessed by multiple variable logistic regression (table 2)

The median age of death was 84 years in the PD group and 59 years in the DAMA group (p<0.0001 (Mann-Whitney U test)) (table 1). Kaplan-Meier curves with stratification by age group, sex and DAMA support this finding, showing separation of survival curves of DAMA versus PD, with a survival advantage of planned discharge in patients under the age of 33.3 and a survival advantage of DAMA in patients over the age of 66.6 years. The stratified Kaplan Meier plots highlight that this survival advantage was restricted to those patients aged over 66.6 years on admission (figure 2). In the younger patients, survival curves separated 2–5 years following discharge (figure 2). The unadjusted Cox proportional hazards model suggested that DAMA offered a survival advantage (HR 2.72, 95%, CI 2.14 to 3.45). Following adjusting for age, sex and SIMD and including an interaction term between age and mode of discharge (to account for the fact that the risk of mortality associated with DAMA reduced with age), Cox proportional hazards models showed DAMA significantly associated with increased risk of death in the youngest patient group (aHR 2.6, 95% CI 1.2 to 5.8) but was protective in the older age group (aHR 0.19, 95% CI 0.07 to 0.49 (table 3).

The odds of death from mental illness and substance misuse, unintentional injuries and those awaiting investigation were all significantly more likely in the DAMA cohort (table 4). These held true in after adjusting for age, sex, length of stay, specialty and 30-day readmission.

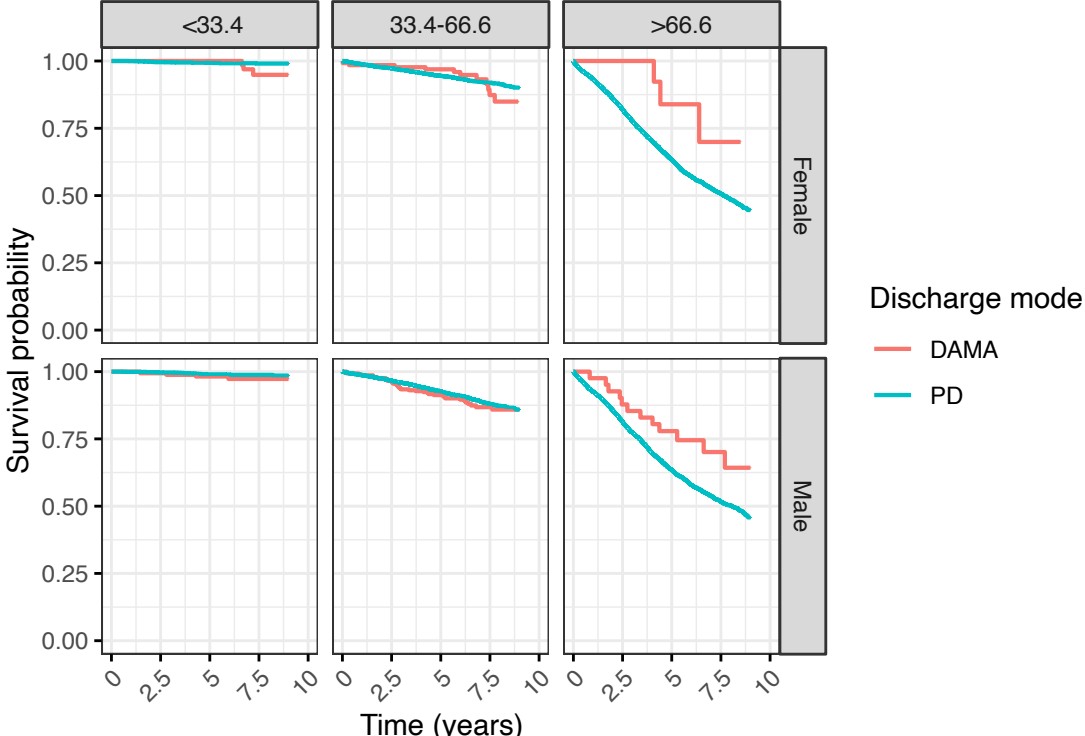

**Figure 2** Kaplan-Meier survival curves broken down by age group and sex. Red lines show planned discharges and green lines show those against medical advice. DAMA, discharge against medical advice; PD, planned discharge.

**Table 3** Cox proportional hazards models for HR of death

| Characteristic | Simple (univariable) models | | | Multiple variable model | | | Full model with interaction | | |
|---|---|---|---|---|---|---|---|---|---|
| | HR | 95% CI | P value | HR | 95% CI | P value | HR | 95% CI | P value |
| Discharge mode | | | | | | | | | |
| Planned discharge | — | — | | — | — | | — | — | |
| DAMA | 0.36 | 0.29, 0.47 | <0.001 | 0.97 | 0.76,1.24 | 0.8 | 2.60 | 1.17, 5.78 | 0.019 |
| Stratified age (years) | | | | | | | | | |
| 13–33.3 | — | — | | — | — | | — | — | |
| 33.4–66.6 | 8.76 | 6.64, 11.5 | <0.001 | 8.97 | 6.77, 11.9 | <0.001 | 9.72 | 7.18, 13.1 | <0.001 |
| >66.6 | 55.5 | 42.4, 72.8 | <0.001 | 58.9 | 44.7, 77.6 | <0.001 | 64.6 | 48.0, 86.9 | <0.001 |
| Sex | | | | | | | | | |
| Female | — | — | | — | — | | — | — | |
| Male | 0.92 | 0.88, 0.97 | <0.001 | 1.04 | 0.98, 1.09 | 0.2 | 1.04 | 0.99, 1.09 | 0.2 |
| Stratified SIMD | | | | | | | | | |
| <4 most deprived | — | — | | — | — | | — | — | |
| ≥4 least deprived | 1.23 | 1.16, 1.30 | <0.001 | 0.87 | 0.83, 0.92 | <0.001 | 0.87 | 0.83, 0.92 | <0.001 |
| Interaction term | | | | | | | | | |
| DAMA * age 33.3–66.6 | | | | | | | 0.48 | 0.20, 1.12 | 0.091 |
| DAMA * age >66.6 | | | | | | | 0.19 | 0.07, 0.49 | <0.001 |

DAMA, discharge against medical advice; SIMD, Scottish Index of Multiple Deprivation.

We assessed how DAMA affected risk of readmission within 30 days of discharge. The indirect standardised incidence rate for 30-day readmission (adjusting for age, sex, SIMD, discharge specialty, month and year of discharge and length of stay) was 1.9 (95% CI 1.5 to 2.2). 30-day readmission was more likely in patients under the age of 66.6 years, those who stayed in hospital <2 days at index admission, who were male, who were discharged from the toxicology unit and who were of more deprived SIMD although difference within strata was only significantly different for SIMD and discharge specialty (figure 3).

## DISCUSSION

This study shows that in a UK-based, unselected, acute medical take, DAMA is more common in younger, male patients of more deprived SIMD. DAMA was associated with increased 30-day readmission rates. Surprisingly, DAMA was associated with an overall reduced mortality. However, following adjustment for sex, age, SIMD and including an interaction term accounting for the reducing mortality seen in older patients who DAMA, we show that patients under the age of 33.3 years who DAMA have an adjusted HR of 2.6 for death while there is a survival advantage for those aged over 66.6 years who DAMA.

This study is the first to demonstrate a reduced HR for death associated with DAMA. The interaction between age and discharge mode age helps to rationalise this result as the reduced mortality was restricted to those patients over 66.6 years. Several factors may account for this finding: Older patients are likely more comorbid and it may be more difficult for them to physically leave without assistance, thus those that were able to DAMA were possibly relatively less comorbid than their counterparts with planned discharges. The support networks in place for the older patients who discharged against medical advice may also have been stronger than in the younger patients. The rate of DAMA in patients over 66.6 years was also much less than that seen in the younger cohort (<33.3 years 5.8%; ≥66.6 years 0.5%) which may have selected out a specific more empowered, less frail population although we have been unable to test this hypothesis with our dataset.

The factors associated with DAMA were consistent with literature (from other healthcare systems)—young, male and from a more deprived background.[1 5 7 8 15–18] While our study did not examine ethnic factors, this was noted to be a risk factor in other studies.[1 5 19 20] It is unclear whether this is a unique identifier for risk of DAMA, or if there is an interaction with other socioeconomic factors. In other healthcare systems, being uninsured has been associated with risk of DAMA; however, we show similar rates of DAMA in a free-at-the-point-of delivery suggesting this factor is potentially less influential.[9 21]

Our study demonstrates a 1.9 times increased risk of 30-day readmission in patients who DAMA. Existing literature suggests that DAMA is associated with increased hospital readmission rate with a range of 1.25–7 times that of the PD population depending on the specific population and presentation.[9 13 15 16 19]

This is the first study of which we are aware examining DAMA in an unselected, medical cohort from the UK. We

**Table 4** Logistic regression of causes of death and their association with discharge against medical advice

| Cause of death | OR | 95% CI | P value |
|---|---|---|---|
| Respiratory infections (pneumonia, all viral respiratory infection including COVID-19, aspiration pneumonia/pneumonitis) | −0.23 | −1.37, 1.24 | 0.71 |
| Malignant neoplasm (all) | 0.27 | −0.92, 1.76 | 0.69 |
| Endocrine, blood and immune disorders (excluding lymphoma/leukaemia) | −12.90 | −264.74, 15.4 | 0.99 |
| Mental and substance use disorders | 3.97 | 0.76, 6.61 | **0.003** |
| Neurological conditions (including dementia, delirium, hypoxic brain injury) | 0.35 | −0.99, 1.91 | 0.62 |
| Cardiovascular diseases (including endocarditis, stroke, bowel ischaemia, pulmonary embolus, bleeding unspecified) | −0.42 | −1.63, 1.09 | 0.53 |
| Respiratory diseases including infective exacerbation of Chronic obstructive pulmonary disease and respiratory failure (excluding malignancy and other respiratory infection) | 0.27 | −1.42, 1.96 | 0.74 |
| Digestive diseases including gastrointestinal bleed, biliary sepsis, cholecystitis (excluding bowel malignancy and ischaemic bowel) | 0.97 | −0.37, 2.52 | 0.18 |

Continued

**Table 4** Continued

| Cause of death | OR | 95% CI | P value |
|---|---|---|---|
| Genitourinary diseases (all renal failure & urinary tract disorder except infection) | 1.01 | −0.68, 2.71 | 0.22 |
| Skin diseases including telangiectasia | −12.89 | NA | 0.99 |
| Musculoskeletal diseases (excluding infection) | −12.89 | −629.78, 56.5 | 0.99 |
| Unintentional injuries (burn, falls, poisoning) | 3.28 | 1.50, 5.05 | **<0.001** |
| Intentional injuries | −12.89 | −576.03, 50.45 | 0.99 |
| Old age, frailty, general decline | −0.79 | −2.82, 1.01 | 0.39 |
| Multiorgan failure | 1.59 | 0.06, 3.23 | 0.04 |
| Death pending investigation, unascertained | 2.78 | 1.33, 4.39 | **<0.001** |

Bold text indicates causes of death in patients who discharged against medical advice with significance p<0.05.

examined a large cohort over a 5-year period providing a robust analysis over an extended time frame. This is also the first study to examine DAMA in a free-at-the-point-of-delivery healthcare setting. It shows that despite free access to healthcare, 3% discharges still occur against medical advice. The temporality of the mortality rate in younger patients who DAMA, highlights that mortality rates were not noticeably different from the PD population until 2.5–5 years following discharge. This suggests that the morbidity associated with the index admission may not directly relate to the mode of death, but likely reflects a more vulnerable patient population.

There are several limitations to our study. We looked at only local, single-centre data for readmission, meaning that if patients were readmitted to other hospitals their data were missed. We also have not been able to accurately assess comorbidity at point of discharge, ethnicity, or how the severity of the presentation specifically affected risk of DAMA, mortality, or risk of readmission.

The younger patients who discharged against medical advice died from largely preventable causes such as substance misuse, mental health disorder and physical harm/injury. A more robust support system is clearly required for these vulnerable patients. Improved follow-up, whether taking the shape of a primary or secondary care physician, psychiatrist, social worker or substance misuse team (but ideally as a coordinated multi-disciplinary team), may provide a structure to identify and manage risk. Training and increasing awareness

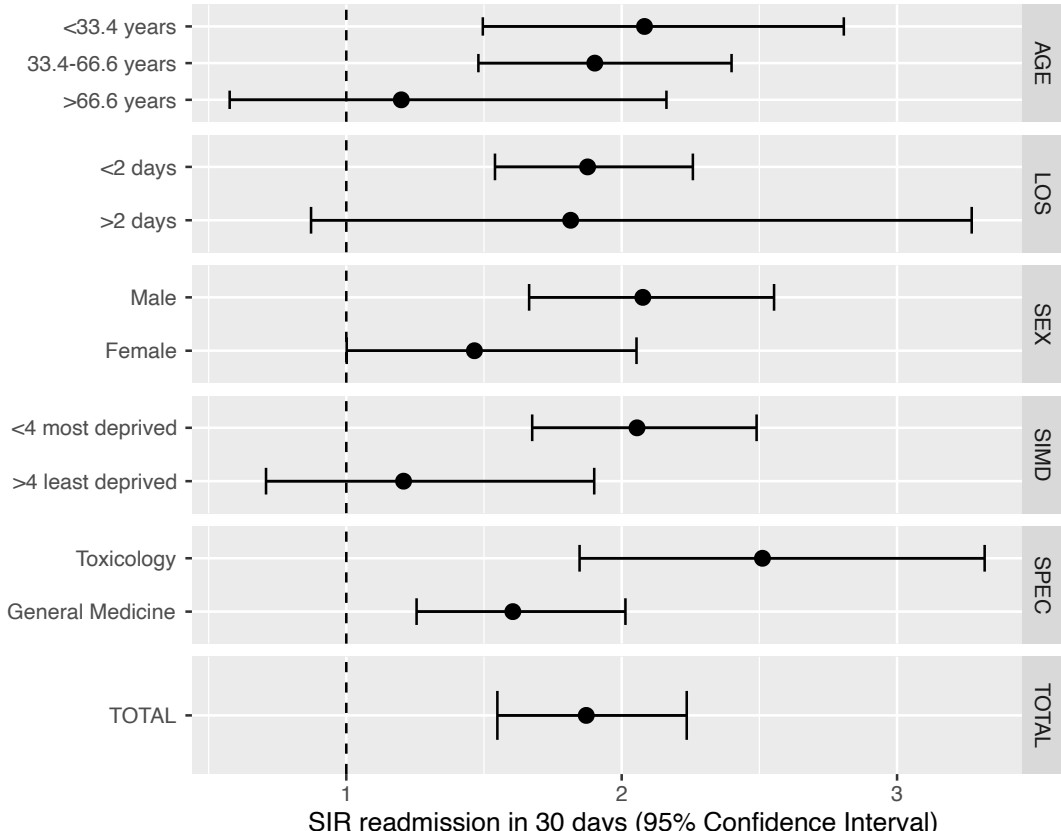

**Figure 3** Standardised incidence rate of 30-day readmission to hospital in DAMA cohort vs planned discharge cohort adjusting for age, length of stay, sex, SIMD, discharge specialty and month and year of discharge. DAMA, discharge against medical advice; LOS, length of stay; SIMD, Scottish Index of Multiple Deprivation; SIR, standardised incidence rate; SPEC, hospital speciality.

of the risks and context surrounding the self-discharging patient, especially among doctors-in-training and nursing staff may help to provide a framework for difficult conversations, appropriate escalation, referral for follow-up (both to deal with the presentation and psycho-social adversity) and debrief and reflection. Significant amounts of anxiety and frustration for both patients and staff often surround situations where patients elect to DAMA. While often framed in a negative way, qualitative research has highlighted the strong empathy felt by the discharging team and families.[22] Framing DAMA by focussing on patient empowerment and autonomy may help to guide hospital policies to focus on a patient-centred approach to DAMA, encouraging shared decision making and safe follow-up planning. Such systemic and cultural changes will likely be necessary to reduce harm for this vulnerable population of patients.

**Author affiliations**
[1]Acute & General Medicine, Royal Infirmary of Edinburgh, Edinburgh, UK
[2]Centre for Discovery Brain Sciences, The University of Edinburgh, Edinburgh, UK
[3]Metabolic Unit, Western General Hospital, Edinburgh, UK
[4]Lothian Analytical Services, Waverley Gate, NHS Lothian, Edinburgh, UK
[5]Department of Acute Medicine, The University of Edinburgh, Edinburgh, UK

**Contributors** AA: Responsible for concept and design of the work, acquisition, analysis and interpretation of data. TC: Responsible for concept and design of the work, acquisition, analysis and interpretation of data. EB: Responsible for acquisition, analysis and interpretation of data. SMG: Responsible for concept and design of the work, analysis and interpretation of data. KAL: Responsible for concept and design of the work, acquisition, analysis and interpretation of data. All authors have been involved in drafting and revision of article and are guarantors of the article and give final approval of the version to be published. They agree to be accountable for all aspects of the work in ensuring that questions related to accuracy or integrity are appropriately investigated and resolved. KAL is guarantor for overall content and accepts full responsibility for the finished work and conduct of the study.

**Funding** This work was supported by the Wellcome Trust-University of Edinburgh Institutional Support Fund (TJGC).

**Competing interests** None declared.

**Patient and public involvement** Patients and/or the public were not involved in the design, or conduct, or reporting, or dissemination plans of this research.

**Patient consent for publication** Not applicable.

**Ethics approval** This work has received Ethics approval from the local Caldicott Guardian Ethics Committee—Reference 17103. Participants gave informed consent to participate in the study before taking part.

**Provenance and peer review** Not commissioned; externally peer reviewed.

**Data availability statement** No data are available.

**ORCID iD**
Anand Alagappan http://orcid.org/0000-0002-0676-0006

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
