## [Reviewer comments · BMJ Open]

ARTICLE DETAILS

TITLE (PROVISIONAL)	How does discharge against medical advice affect risk of mortality and unplanned readmission? A retrospective cohort study set in a large UK medical admissions unit.
AUTHORS	Alagappan, Anand; Chambers, Tom; Brown, Erik; Grecian, Sheila; Lockman, Khalida Ann

VERSION 1 – REVIEW

REVIEWER	Kemp, Kyle University of Calgary, Community Health Sciences
REVIEW RETURNED	05-Nov-2022

GENERAL COMMENTS	Thank you for the opportunity to review this extremely well-written paper which examined the potential associations of being discharged against medical advice (DAMA), mortality, and 30-day unplanned readmissions. In my opinion, this is an excellent paper with important findings. My specific comments are as follows: Although somewhat intuitive, please include that all readmissions were unplanned ones, where mentioned in the abstract. The introduction is excellent, and provides a strong rationale for the research in the context of limitations of prior work. The methods section is easy to follow. One potential addition (if available) may be to look at the potential association between DAMA and follow-up visits to an emergency department. Given the findings relative to mental and substance abuse disorders, this cohort may present to an emergency department, without a resulting admission to hospital. When collapsing continuous variables into categories, please provide a rationale with supporting evidence (if available). For example, the age group extending to 33.3 years seems arbitrary. In the results section, it may be worthwhile to provide a flow diagram to reflect the numbers provided in the first paragraph well-written. It provides a sound rationale for the research and orients the reader to the context of the study. The discussion is well-written and the accompanying tables and figures are appropriate.
--

	Thank you one again for the opportunity to review this paper, which I very much enjoyed reading.
--	--

REVIEWER	Warriner, David
REVIEW RETURNED	07-Nov-2022

GENERAL COMMENTS	This is without doubt one of the most well written papers I've ever had the pleasure of peer reviewing. Precious little to criticise - a simple and elegant question answered in a comprehensive and robust manner. The authors are aware of the few limitations of the study, but even the 3% DAMA figure itself, which to my knowledge (after being active in this area for over a decade) is the first time in the UK we have had such data, is worthy of publication alone. Whilst many of the findings are indeed unsurprising and expected from other studies across the globe e.g. deprivation, gender, age, readmission rates, again this is the first time it has been demonstrated in a UK cohort. The reduced HR for death with DAMA in those age 66.6 is an interesting finding, not seen before, and whilst it is certainly hypothesis generating I think it would require further replication across populations before firm conclusions were made. Finally, NHS Digital doesn't currently capture such DAMA data explicitly and as a result this paper will be impactful and well cited in years to come I am sure. Many thanks.
---

REVIEWER	Chivite, David Hospital Universitari de Bellvitge, Internal Medicine
REVIEW RETURNED	16-Nov-2022

GENERAL COMMENTS	The paper submitted for review explores in a retrospective fashion the circumstances and outcomes of discharge against medical advice (DAMA) among a sizable cohort of Scottish patients admitted for a variety of "medical" reasons to acute care. The main findings are a) DAMA patients are younger, male and with more social deprivation and b) younger DAMA patients are at risk of increased mortality in the mid- to long-term. The study addresses a not so common but relevant problem, given the association found in prior and the present paper between DAMA and subsequent medical risks. It provides a clear picture of the problem in a "free", public health system, thus avoiding the bias related to direct medical costs & financial constraints for the patient in other health care settings. The design is retrospective and thus some key data on events related to a greater likelihood of DAMA during the index admission are missing, but measures of social deprivation and the type of admission ("general" vs "toxicology") capture well known risk factors for poor outcomes. The aims of the study are clear, data collection is thorough and the statistical analysis is comprehensive, thus allowing the authors to progress from the crude first analysis to the final adjusted model that allows for the finding of the higher risks among the younger, more deprived patients. The paper can however be easily read and the discussion and conclusions are clear and relate fully to the findings
---

	All that said I think that this paper might be accepted with no further revisions A few remarks on very basic issues: The first mention to DAMA in the abstract is not preceded by the words making up the acronym. There is also an acronym (MSK, likely "musculoskeletal") unexplained in the table on page 11
--	--

VERSION 1 – AUTHOR RESPONSE

Reviewer: 1

Dr. Kyle Kemp, University of Calgary, Alberta Health Services

Comments to the Author:

Thank you for the opportunity to review this extremely well-written paper which examined the potential associations of being discharged against medical advice (DAMA), mortality, and 30-day unplanned readmissions.

In my opinion, this is an excellent paper with important findings.

My specific comments are as follows:

Although somewhat intuitive, please include that all readmissions were unplanned ones, where mentioned in the abstract.

- We clarify that we only included unscheduled admissions in our re-admission analysis. We have modified the abstract to confirm this.

The introduction is excellent, and provides a strong rationale for the research in the context of limitations of prior work.

The methods section is easy to follow. One potential addition (if available) may be to look at the potential association between DAMA and follow-up visits to an emergency department. Given the findings relative to mental and substance abuse disorders, this cohort may present to an emergency department, without a resulting admission to hospital.

- We agree with reviewer one that review of number A and E attendances (before and after index presentation) would be another interesting analysis to pursue but our analytics team are unable to provide this information in time for the deadline set for response.

When collapsing continuous variables into categories, please provide a rationale with supporting evidence (if available). For example, the age group extending to 33.3 years seems arbitrary.

- We have provided rationale for definition of our age strata. As standardised mortality rates crossed unity at the age of 36 years, we defined three strata below, around and above this intersection. We divided age in years at index presentation by 33.3 to achieve the three age groups we used for analyses.

In the results section, it may be worthwhile to provide a flow diagram to reflect the numbers provided

in the first paragraph well-written. It provides a sound rationale for the research and orients the reader to the context of the study.

- We have provided a flow diagram for patients included and excluded from analyses as a new Figure 1.

The discussion is well-written and the accompanying tables and figures are appropriate.

Thank you one again for the opportunity to review this paper, which I very much enjoyed reading.

Reviewer: 2

David Warriner

Comments to the Author:

This is without doubt one of the most well written papers I've ever had the pleasure of peer reviewing. Precious little to criticise - a simple and elegant question answered in a comprehensive and robust manner. The authors are aware of the few limitations of the study, but even the 3% DAMA figure itself, which to my knowledge (after being active in this area for over a decade) is the first time in the UK we have had such data, is worthy of publication alone. Whilst many of the findings are indeed unsurprising and expected from other studies across the globe e.g. deprivation, gender, age, readmission rates, again this is the first time it has been demonstrated in a UK cohort. The reduced HR for death with DAMA in those age 66.6 is an interesting finding, not seen before, and whilst it is certainly hypothesis generating I think it would require further replication across populations before firm conclusions were made. Finally, NHS Digital doesn't currently capture such DAMA data explicitly and as a result this paper will be impactful and well cited in years to come I am sure. Many thanks.

Reviewer: 3

Dr. David Chivite, Hospital Universitari de Bellvitge

Comments to the Author:

The paper submitted for review explores in a retrospective fashion the circumstances and outcomes of discharge against medical advice (DAMA) among a sizable cohort of Scottish patients admitted for a variety of "medical" reasons to acute care. The main findings are a) DAMA patients are younger, male and with more social deprivation and b) younger DAMA patients are at risk of increased mortality in the mid- to long-term.

The study addresses a not so common but relevant problem, given the association found in prior and the present paper between DAMA and subsequent medical risks. It provides a clear picture of the problem in a "free", public health system, thus avoiding the bias related to direct medical costs & financial constraints for the patient in other health care settings. The design is retrospective and thus some key data on events related to a greater likelihood of DAMA during the index admission are missing, but measures of social deprivation and the type of admission ("general" vs "toxicology") capture well known risk factors for poor outcomes.

The aims of the study are clear, data collection is thorough and the statistical analysis is comprehensive, thus allowing the authors to progress from the crude first analysis to the final adjusted model that allows for the finding of the higher risks among the younger, more deprived patients. The paper can however be easily read and the discussion and conclusions are clear and relate fully to the findings

All that said I think that this paper might be accepted with no further revisions

A few remarks on very basic issues: The first mention to DAMA in the abstract is not preceded by the words making up the acronym. There is also an acronym (MSK, likely "musculoskeletal") unexplained in the table on page 11

- We have reviewed the abbreviations to ensure they are clear at first use.

VERSION 2 – REVIEW

REVIEWER	Kemp, Kyle University of Calgary, Community Health Sciences
REVIEW RETURNED	27-Dec-2022

GENERAL COMMENTS	Thank you for addressing the comments raised during our initial review. Once again, my kudos to the authors on a very well-written and relevant paper. I have no further commentary and look forward to seeing this work in the Journal.
--